# Synthesis of New Derivatives of Benzofuran as Potential Anticancer Agents

**DOI:** 10.3390/molecules24081529

**Published:** 2019-04-18

**Authors:** Mariola Napiórkowska, Marcin Cieślak, Julia Kaźmierczak-Barańska, Karolina Królewska-Golińska, Barbara Nawrot

**Affiliations:** 1Chair and Department of Biochemistry, Medical University of Warsaw, 1 Banacha Str., 02-097 Warsaw, Poland; 2Centre of Molecular and Macromolecular Studies, Polish Academy of Sciences, 112 Sienkiewicza Str., 90-363 Łódź, Poland; julia@cbmm.lodz.pl (J.K.-B.); kkrolews@cbmm.lodz.pl (K.K.-G.); bnawrot@cbmm.lodz.pl (B.N.)

**Keywords:** benzofurans, chemical synthesis, cytotoxic properties, HeLa, MOLT-4, K562

## Abstract

The results of our previous research indicated that some derivatives of benzofurans, particularly halogeno-derivatives, are selectively toxic towards human leukemia cells. Continuing our work with this group of compounds we here report new data on the synthesis as well as regarding the physico-chemical and biological characterization of fourteen new derivatives of benzofurans, including six brominated compounds. The structures of all new compounds were established by spectroscopic methods (^1^H- and, ^13^C-NMR, ESI MS), and elemental analyses. Their cytotoxicity was evaluated against K562 (leukemia), MOLT-4 (leukemia), HeLa (cervix carcinoma), and normal cells (HUVEC). Five compounds (**1c**, **1e**, **2d**, **3a**, **3d**) showed significant cytotoxic activity against all tested cell lines and selectivity for cancer cell lines. The SAR analysis (structure-activity relationship analysis) indicated that the presence of bromine introduced to a methyl or acetyl group that was attached to the benzofuran system increased their cytotoxicity both in normal and cancer cells.

## 1. Introduction

Benzofuran skeleton holds an important position in organic chemistry and it is considered to be one of the most important heterocyclic systems because of its diverse profile of biological activity. This structural unit is a central part of a variety of biologically active compounds. Natural and synthetic benzofuran derivatives have been reported to possess wide therapeutic properties, including antiviral, immunosuppressive, antioxidant, antifungal, anti-inflammatory, antimicrobial, analgesic, antihyperglycemic, and antitumor activities [1,2,3,4,5,6]. *Cicerfuran, Conocarpan*, and *Ailanthoidol* are the best known biologically active natural benzofurans (Figure 1). Specifically, the *Cicerfuran* shows antifungal activity, *Conocarpan* has been reported as an antifungal and antitrypanosomal agent, and *Ailanthoidol* exhibits anticancer, antiviral, immunosuppressive, antioxidant, and antifungal activity [1,2,7]. The synthetic benzofuran derivatives are represented by *Amiodarone* (Figure 1), being used in the treatment of ventricular and supraventricular arrhythmias, and by *Bufuralol*, which is a non-specific β-adrenergic blocker with an affinity for β1 and β2-adrenergic receptors [3,4,7]. 

Nowadays, when cancer, after cardiovascular diseases, is the second most common cause of death and still constitutes an unresolved problem of clinical medicine and pharmacology, extensive research regarding new anticancer compounds is especially important. These new drugs should possess improved pharmacokinetics and specifically destroy cancer cells, without causing negative side effects. Research in the group of benzofuran derivatives is justified, especially by the fact that one can find many examples of data in the literature on benzofurans with anticancer activity. In many cases, the benzofuran skeleton is fused with other heterocyclic or aromatic moieties (Figure 2). 

There are several benzofuranyl imidazole derivatives among them (**I** and **II**), which were found to be cytotoxic towards an ovarian carcinoma cell line (Skov-3). The study of *N*-(5-(2-bromobenzyl)thiazole-2-yl) benzofuran-2-carboxamide (**III**) showed that this compound inhibited the growth of HCC (human hepatocellular carcinoma) cells and induced their apoptosis. The synthetic derivative **IV** was found to have antitumor activity and it was an effective chemopreventive and chemotherapeutic agent against malignant T cells. Moreover, a series of triazole derivatives **V** showed moderate antitumor activity. Recent data reported this type of benzofuran derivatives as potential therapeutic agents for breast cancer [7].

These are only a few examples from a large group of benzofurans with anticancer activity. Moreover, examples of compounds with cytotoxic activity were found among the simple derivatives of benzofuran like 2- and 3-benzofuranocarboxylic acid derivatives **VI**–**VII** (Figure 3). The above-mentioned compounds exhibit significant cytotoxic activity against human cancer cell lines [8].

The literature survey shows that benzofurans containing halogens in their structure constitute an important group of compounds, with antitumor, cytotoxic, spasmolytic, antiarrhythmic, and antifungal activity [9,10,11,12,13,14,15,16,17,18,19,20]. Thus, this group of compounds is interesting for researchers, especially because the presence of halogen can increase the activity and selectivity of derivatives. This is probably related to the ability of halogens to create a “halogen bond”, which results from the formation of the σ-hole. Although the halogen bonds are weaker than hydrogen bonds, they have specific effects and can lead to significant gains in binding affinity. These interactions can be found in protein-receptor complexes as well as in small molecules [21,22,23,24]. 

Furthermore, we have identified three bromo derivatives **VIII**–**X** (Figure 4) that showed selective toxicity towards human leukemia cells in our previous studies [25,26]. Compound **VIII** is especially cytotoxic towards K562 and HL-60 leukemic cell lines (IC_50_ 5.0 and 0.1 µM, respectively), however it is not toxic towards HeLa cancer cells and healthy endothelial cells (HUVEC) (IC_50_ > 1 mM). Moreover, the observed remarkable cytotoxicity of **VIII** towards K562 cells resulted from cells apoptosis. Compounds **IX** and **X** proved to be highly toxic towards cancer cells (IC_50_ in a few µM range) and non-toxic towards endothelial cells (HUVEC) [25,26,27]. Unfortunately, these compounds are poorly soluble in water, which limits their use in the cell culture or animal studies.

Using **VIII**–**X** as the lead compounds, we designed and synthesized fourteen new derivatives with hopes for their better solubility in aqueous solutions (of lower lipophilicity when compared to **VIII**–**X**). The biological activity (i.e., cytotoxicity, activation of apoptosis, interaction with DNA) of these new derivatives was also evaluated. 

## 2. Results

### 2.1. Synthesis

Our goal was to obtain a small library of new, less lipophilic derivatives/analogs of lead compounds **VIII**–**X**. We designed the synthesis of a set of compounds containing a carboxyl (**1**), formamide (**1a**), and methoxycarbonyl groups (**1b**), instead of an acetyl group in the position 2 of the parent benzofuran ring to obtain new benzofuran **VIII** analogs. 

Thus, the starting acid **1**, which was obtained by the multistep synthesis according to the previously reported procedures [28] was submitted either to oxalyl chloride and ammonium solution treatment or methylated with dimethyl sulphate, delivering the amide derivative **1a** and methyl ester **1b**, respectively (Scheme 1). In the next step, compounds **1** and **1b** were submitted to bromination. For this purpose, ester **1b** was reacted with molecular bromine in chloroform. Under these conditions, hydrogen in the methyl group at position 3 was substituted by a bromine atom to give compound **1c**, which only differed by the substituent in position 2 (methoxycarbonyl versus acetyl). During bromination of the acid **1** using bromine in chloroform or NBS in CCl_4,_ a mixture of products was obtained, which was difficult to separate. Thus, the reaction conditions were changed and ethanol, instead of CCl_4,_ was used as a solvent in bromination reaction that was carried out in the presence of the NBS, while acetic acid was used as a solvent in the respective reaction that was carried out in the presence of bromine. Under these conditions, we managed to isolate the bromo-derivative **1d**, with satisfactory yield. Moreover, bromo-derivative **1e** was also obtained, but only in the reaction that was facilitated by NBS. The analyses of nuclear magnetic resonance spectra (^1^H- and ^13^C-NMR), mass spectra, and elemental analysis showed that the structures of the received compounds were different from the assumed ones (bromination of the methyl group at position 3). Instead, the derivatives in which the carboxyl group was replaced by the bromine atom at the position 2 were isolated. Moreover, we confirmed the formation of **1e**, in which one bromine atom substituted the hydrogen atom in the acetyl group of the benzene moiety of benzofuran ring. The use of polar protic solvents (acetic acid, ethanol) could explain this substitution.

Analysis of the calculated log*P* values (Table 1) has shown that carboxyl and formamide analogs of lead compound **VIII** (**1** and **1a**, respectively) are much less lipophilic, while methoxycarbonyl analogs **1b** and **1c** exhibit similar properties as **VIII**. In contrast, an introduction of the bromine atom at position 2 of the furane ring resulted in significant increase of the benzofuran system hydrophobicity. 

Scheme 2 and Scheme 3 show the syntheses of analogs of compounds **IX** and **X**, respectively. The starting material in both cases was 6-acetyl-5-hydroxy-2-methylbenzofuran-3-carboxylic acid (**2**), which was subjected to multidirectional transformations (Scheme 2). In the first approach, the amide-derivative **2a** was obtained in the reaction of the acid **2** with oxalyl chloride and ammonium hydroxide. Next, the bromo-derivative **2b** was obtained by the bromination of **2a** with bromine in acetic acid as a solvent. The nuclear magnetic resonance (^1^H- and ^13^C-NMR), mass spectrometry and elemental analyses confirmed the substitution of hydrogen by bromine in aromatic ring in the position 4. We assume that the presence of the OH group in position 5 of the benzene ring assisted in the electrophilic substitution of bromide cation in its ortho position. In the second path, the acid **2** was brominated in the same conditions and the bromo-derivative **2e**, also with a bromine substitution to the benzene ring, was obtained. In the third approach, an ester-derivative **2c**, which was obtained in the reaction of the acid **2** with dimethyl sulphate, was brominated by using NBS in CCl_4_ to give the derivative **2d**, with a bromomethyl group in the position 2. Interestingly, all of the obtained derivatives **2** (**2a**–**2e**) exhibited lower clog*P* values, confirming the better water solubility of derivatives compared to lead compound **IX**, and the most pronounced occurred primary carboxamides **2a** and **2b**. 

To obtain analogs of compound **X**, the starting acid **2** was reacted with an excess of dimethyl sulphate and the obtained derivative **3** was subjected to a multidirectional synthesis (Scheme 3). In the first case, bromine was introduced into the methyl group to give a compound **3d**, by reaction with NBS in CCl_4_. In the reaction of the compound **3** with a bromine in acetic acid, the lead compound **X** was obtained and then finally reduced to provide a hydroxyl-derivative **3a**. In the third path, ester **3** was hydrolyzed in alkaline conditions to acid **3b**, and finally this derivative was converted to an amide **3c** by reaction with oxalyl chloride and ammonium hydroxide. Importantly, all of the new benzofuran derivatives related to **X** are characterized by lower clog*P* values when compared to the lead compound, indicating their improved solubility in aqueous media (Table 1).

The introduced substituents significantly affected the lipophilicity of the obtained benzofurans. In most cases, the new derivatives had a lower clog*P* value in comparison with the lead compounds. The exception are three derivatives **1c**, **1d**, **1e**, where the substitution of bromine in the furan ring (compounds **1d** and **1e**) or in the methyl group caused the clog*P* to increase (Table 2).

### 2.2. MTT Cytotoxicity Studies

Fourteen new benzofuran derivatives were tested for their cytotoxic properties in K562, MOLT-4 (leukemia, suspension cells), HeLa (cervix carcinoma, adherent cells), and normal endothelial cells (HUVEC). First, we measured the viability of cells after 48 h incubation with the given compound at the concentration of 100 μM. Next, for compounds that reduced cancer cells survival for more than 50%, we determined IC_50_ values. Cells that were exposed to 1% DMSO (a vehicle) served as the control with 100% survival. Cells treated with 1 µM staurosporine served as the internal control of the cytotoxicity experiments.

We have identified five compounds **1c**, **1e**, **2d**, **3a**, and **3d** in the initial screening, which, at the concentration of 100 μM, reduced the viability of all tested cancer cells K562, HeLa and MOLT-4 for more than 50% (data not shown). These compounds were also cytotoxic against human normal endothelial cells, so these compounds did not show any selectivity between the cancer and normal cells.

Next, for compounds **1c**, **1e**, **2d**, **3a**, and **3d**, the IC_50_ values were calculated (data given in Table 2). The test compounds were similarly toxic toward both cancer and normal cells, with IC_50_ values in the range of 20–85 μM. The exceptions were compounds **1c** and **3d**, which show IC_50_ out of this range (180 μM for MOLT-4 and 6 μM for HUVEC cells, respectively). We did not observe any significant differences in susceptibility between adherent and suspension cell lines. 

For compounds **1c**, **1e**, **2d**, and **3d**, that exhibited the highest toxicity for K562 leukemia cells (IC_50_ below 50 µM), we investigated whether they induce apoptosis in these cells. We have measured the activity of caspases 3 and 7 (caspase 3/7), which are markers of programmed cell death. The K562 cells were treated with 1% DMSO (negative control), 1 µM staurosporine (positive control), or a test compound at the concentration of 5 × IC_50_ for 18 h. The activity of caspase 3/7 was measured using pro-fluorescent peptide substrate. 

As shown in Figure 5, staurosporine, which is a strong inducer of apoptosis significantly increased the activity of caspase 3/7 in K562 cells. On the other hand, 1% DMSO had no effect on the activation of caspases. Interestingly, the incubation of cells with compound **1e** resulted in nearly five-fold increase in the activity of caspase 3/7, while compounds **1c** and **2d** activated caspase 3/7 to a lesser extent (about two-fold increase). In the presence of compound **3d**, the activation of caspases was minimal, if any. Altogether, this result suggests that the cytotoxic activity of test benzofurans **1c**, **1e**, and **2d** in K562 cells may be due to the induction of death by apoptosis.

### 2.3. Interaction with DNA

The results of MTT cytotoxicity experiments indicated that compounds **1c**, **1e**, **2d**, and **3d** were highly toxic towards the used cell lines. We hypothesized that a possible explanation of observed cytotoxicity might be due to an interaction of test benzofurans with genomic DNA (e.g., by intercalation). To verify this hypothesis, we investigated whether the test benzofurans have any effect on digestion of a plasmid DNA (pcDNA3.1 HisC) with endonuclease *BamH1*. pcDNA3.1 HisC contains a unique *BamH1* restriction site which allows for plasmid linearization. Plasmid DNA exists in linear, superhelical, and circular forms that differ in electrophoretic mobility (Figure 6 lane 1). Plasmid DNA was converted to a linear form upon digestion with *BamH1* (lane 2). Daunorubicin, which is a strong intercalator to double-stranded DNA, was used as a control in this experiment and it completely inhibited the digestion of plasmid DNA with *BamH1* (lane 3). In the presence of test compounds, pcDNA3.1 HisC was partially digested with *Bam H1* restriction enzyme (Figure 6 lanes 4–7). Most of the plasmid DNA was converted to a linear form, however there is a circular form still present. These results suggest that test benzofurans, to some extent, interact with DNA (especially compounds **1c**, **1e**, **2d**), and this interaction inhibits the digestion of double stranded DNA chain with restriction endonuclease.

## 3. Discussion

We have previously identified benzofurans **VIII**, **IX**, and **X** (lead benzofurans), which efficiently killed cancer cells and were not toxic toward normal endothelial cells [25,26,27]. Moreover, lead compound **VIII** demonstrated selective toxicity toward leukemia cell lines. However, these compounds were poorly soluble in aqueous solutions. Based on their structure, we have synthesized 14 new derivatives with decreased lipophilicity. The polarity of new compounds was predicted based on the calculated log*P* values. We tested their cytotoxic properties in human cells of cancer and normal origin. Five compounds, **1c**, **1e**, **2d**, **3a**, and **3d**, displayed the highest cytotoxicity toward cancer cell lines. However, these compounds were less toxic than lead compounds **VIII**, **IX**, **X**, and did not demonstrate any selectivity toward leukemia cells (Table 2). Moreover, new derivatives exhibited significant toxicity in normal endothelial cells. Cells death is usually carried out in one of the two major mechanisms: apoptosis or necrosis. Apoptosis is a highly regulated and controlled process that does not elicit an inflammation response at the site of cell death. Necrosis leads to sudden and uncontrolled cell disintegration that is associated with release of the cellular content and massive inflammation [29]. Therefore, in the next experiments, we investigated whether the cellular toxicity of new benzofurans is the result of apoptosis or necrosis. Our data demonstrate that the activity of caspase 3/7 (an apoptosis marker) is significantly increased (1.5- to 5-fold) in the presence of benzofurans (Figure 5). It suggests that these derivatives induce apoptosis in cancer cells. In the search of cellular targets for testing benzofurans, we examined whether DNA may be such a target. Using biochemical assay, we found out that the incubation of test compounds with plasmid DNA inhibited its cleavage with selected endonuclease (*BamH1*). A similar result was obtained with daunorubicin, which is a strong DNA intercalating agent. The presence of undigested plasmid DNA suggests that benzofurans intercalate to DNA (or bind DNA in other way). However, a comparison of DNA digestion products clearly indicates that the binding of benzofurans to DNA is much weaker than daunorubicin (Figure 6). 

The presence of a bromine substituent in the alkyl chain attached to the furan ring is most likely to be responsible for cytotoxic activity of compounds **1c**, **2d**, and **3d**. The activity of compound **1e** is probably related to the presence of a bromoacetyl substituent in a benzene ring. Whilst the presence of a halogen (bromine) directly substituted to the benzene ring or the furan skeleton does not seem to increase the cytotoxic activity of the tested compounds, for example, **2e**, **2b**, and **1d**. The amide derivatives of benzofurans (compounds **2a**, **3c**, **1a**) that lack halogen-containing alkyl substituents (a bromine atom) did not show the cytotoxic properties toward cancer cell lines. A similar effect is observed for derivatives with a free acidic group (compounds **1**, **3b**) and ester derivatives (**1b**).

We can observe a marked decrease in the activity and selectivity of these derivatives when comparing the activity of bromo-derivatives **1c** and **1e** with the activity of the lead compound **VIII**. This effect is probably due to the absence of the acetyl group at the 2-position of the furan ring. The derivative **1c** has an ester group and the compound **1e** a bromine atom in this position. Thus, it can be concluded that the arrangement of substituents: the acetyl group at the 2-position and the bromomethyl at the three-position determines the activity and selectivity of the lead compound. 

Analysis of the results for the active **2d** derivative in comparison to its lead compound **IX** also indicates that the structural modifications of **2d** resulted in a loss of selectivity and decreased activity. In this case, the derivatives differ in the location of the halogen atom. The **2d** derivative contains the bromomethyl substituent in the two-position and the acetyl group in the six-position of the benzofuran system, while the **IX** contains the methyl group in the two-position and the bromoacetyl substituent in the six-position of the benzofuran system. It can again be assumed that the presence of a halogen atom substituted to an alkyl/acetyl moiety determines the activity of the derivatives, but the appropriate positioning of substituents is important in their selectivity. Finally, by comparing the active derivatives **3a** and **3d** with their lead compound **X**, we also observe a decrease in activity and selectivity. The **3a** compound differs from the leading compound by the presence of a hydroxyl group. It can be hypothesized that the reduction of the keto group and the possibility of creating additional hydrogen bonds, as well as an increase in the hydrophilicity could affect the activity of this derivative.

Compound **3d** contains a bromomethyl substituent at the two-position and an acetyl group at the six-position, while **X** contains a bromoacetyl group at the six-position and a methyl at the two-position. Both of the compounds exhibit cytotoxicity, but the absence of the bromoacetyl substituent in compound **3d** eliminated its selectivity and decreased cytotoxicity to the cancer cells.

## 4. Materials and Methods

### 4.1. Chemistry

All of the solvents, reagents, and chemicals used in these studies were purchased from Aldrich Chemical (Saint Louis, MO, USA) and Merck AG (Saint Louis, MO, USA). The melting points were determined with Electrothermal 9100 capillary apparatus and they are uncorrected. The nuclear magnetic resonance spectra (University of Warsaw, Warsaw, Poland) were recorded in DMSO-*d*_6_ or CDCl_3_ on VMNRS300 operating at 300 MHz (^1^H-NMR) and 75 MHz (^13^C-NMR). Chemical shifts (δ) are expressed in parts per million relative to tetramethylsilane used as the internal reference. The coupling constants (*J*) values are given in hertz (Hz) and spin multiples are given as s (singlet), d (dublet), t (triplet), and m (multiplet). Mass spectral ESI (Electrospray Ionization) measurements were carried out on a MicrOTOF II, Bruker instrument with a TOF detector (Jagiellonian Univeristy in Krakov, Poland). The spectra were obtained in the positive ion mode. Elemental analyses were recorded with CHNS micro analyzer elementary model Vario Micro Cube with electronic microbalance (Jagiellonian Univeristy in Krakow, Poland). Flash chromatography was performed on Merck Kieselgel 0.05–0.2 mm reinst (70–325 mesh ASTM, Saint Louis, MO, USA) silica gel using chloroform as eluent. TLC monitored progress of the reactions described in the experimental section on silica gel (plates with fluorescent indicator 254 nm, layer thickness 0.2 mm, Kieselgel G. Merck, Saint Louis, MO, USA), using chloroform-methanol as an eluent system at the *v*/*v* ratio of 9.8:0.2 or 9.5:0.5. 

### 4.2. General Synthetic Procedures

Procedure 1. Procedure for Synthesis of Amides.

An appropriate carboxylic acid (0.004 mol) was suspended in anhydrous dichlorometane (DCM) (10 mL). Next, oxalyl chloride (0.43 mL, 0.005 mol) and the one drop of dimethylformamide (DMF) were added to the solution. The reaction mixture was stirred at room temperature for 24 h. Then, ammonium solution (aq. 30%, 5 mL) was added drop by drop and the mixture was stirred at room temperature for additional 12 h. When the reaction was complete (TLC control) the resulting mixture was diluted with water (50 mL) and extracted with DCM (3 × 50 mL). The organic extracts were dried with magnesium sulfate and concentrated under reduced pressure. The resulting solid was purified by a silica gel column chromatography (eluent: chloroform or chloroform:methanol; 50:0.2, *v*/*v*). 

Procedure 2. Procedure for the Preparation of Methyl ester. 

Procedure according to the method described earlier [11]. Thus, a mixture of appropriate carboxylic acid (0.02 mol), K_2_CO_3_ (0.1 mol) and (CH_3_O)_2_SO_2_ (0.02 mol) in acetone was refluxed for 48 h. When the reaction was complete, the mixture was filtered and the solvent was removed on rotary evaporator [11]. The residue was purified by a silica gel column chromatography (eluent: chloroform or chloroform:methanol; 50:0.2 *v*/*v*).

Procedure 3. Procedure for Bromination by Using N-Bromosuccinimide (NBS). 

In this method, the procedure was used, as described earlier [11]. Briefly, *N*-bromosuccinimide (NBS) (0.02 mol) and the catalytic amount of benzoyl peroxide were added to a solution of the appropriate ester or acid (0.02 mol) in dry carbon tetrachloride or alternatively in ethanol (50 mL). The reaction mixture was refluxed for 24 h. When the reaction was complete (TLC monitoring), the mixture was filtered and the solvent was removed under reduced pressure. Silica gel column chromatography purified the residue (eluent: chloroform or chloroform:methanol; 50:0.2 *v*/*v*). 

Procedure 4. Procedure for Bromination by Using Br_2_. 

Method a. Procedure according to the method that was described earlier [11]. Thus, an appropriate ester, amide, or acid (0.02 mol) was dissolved in CHCl_3_ (20 mL), and then a solution of bromine in CHCl_3_ (0.02 mol in 10 mL) was added dropwise with stirring for 1 h. The obtained mixture was stirred at room temperature for 24 h. When the reaction was finished, the solvent was removed under reduced pressure. The residue was purified by a silica gel column chromatography (eluent: chloroform or chloroform:methanol; 50:0.2 *v*/*v*).

Method b. Procedure according to the method described earlier [11]. Thus, an appropriate ester, amide, or acid (0.02 mol) was dissolved in CH_3_COOH (80%, 20 mL), and then a solution of bromine in CH_3_COOH (0.02 mol in 10 mL) was added dropwise with stirring for 1 h. The obtained mixture was stirred at room temperature for 24 h. When the reaction was complete, the resulting mixture was diluted by Na_2_S_2_O_3_ solution (10 mL) and extracted with DCM (3 × 50 mL). The obtained organic extracts were dried with calcium chloride, filtered, and concentrated under reduced pressure. Silica gel column chromatography purified the residue (eluent: chloroform or chloroform:methanol; 50:0.2 *v*/*v*).

Procedure 5. Procedure for Reduction. 

A starting ketone (0.0024 mol) was dissolved in the peroxides-free dioxane (20 mL), and then NABH(OAc)_3_ (0.0048 mol) was added. The mixture was stirred at room temperature for 24–48 h. When the reaction was complete, the solvent was removed under reduced pressure. The solid residue was dissolved in CHCl_3_ (50 mL) and then washed with water (3 × 20 mL). The organic solution was dried with magnesium sulfate, filtered, and concentrated under reduced pressure. Finally, silica gel column chromatography purified the residue (eluent: chloroform or chloroform:methanol; 50:0.2 *v*/*v*).

#### 4.2.1. Synthesis of Analogues of Compound **VIII**

##### Synthesis of 7-Acetyl-5,6-Dimethoxy-3-Methylbenzofuran-2-Carboxylic Acid (**1**)

7-Acetyl-5,6-dimethoxy-3-methylbenzofuran-2-carboxylic acid was obtained in the multistep reaction according to the method described earlier [28].

M.W. = 278.2573; C_14_H_14_O_6_; Yield: 30%; white powder, m.p. 212–214 °C; ^1^H-NMR (300 MHz, DMSO, δ/ppm): 2.50 (3H, s, -CH_3_), 2.62 (3H, s, -COCH_3_), 3.84 (3H, s, -OCH_3_), 3.92 (3H, s, -OCH_3_), 7.47 (1H, s, Ar-H), 13.41 (1H, br.s, -COOH); ^13^C-NMR: δ 9.15, 32.13, 56.43, 61.755, 105.50, 119.64, 124.39, 124.68, 142.19, 144.27, 147.40, 150.07, 160.76, 197.33;.HRMS (*m/z*): calculated value for [M + Na] 100% = 301.0683; found 100% = 301.0681^+^; Anal. Calc. for C_14_H_14_O_6_: 60.43% C; 5.07% H, found 59.25% C; 4.92% H.

##### Synthesis of 7-Acetyl-5,6-Dimethoxy-3-Methylbenzofuran-2-Carboxamide (**1a**)

7-Acetyl-5,6-dimethoxy-3-methylbenzofuran-2-carboxamide was obtained according to Procedure 1.

M.W. = 277.2726; C_14_H_15_NO_5,_ Yield: 37%; white powder, m.p. 203–205 °C; ^1^H-NMR (300 MHz, CDCl_3_, δ/ppm): 2.50 (3H, s, -CH_3_), 2.68 (3H, s, -COCH_3_), 3.80 (3H, s, -OCH_3_), 3.90 (3H, s, -OCH_3_), 7.41 (1H, s, Ar-H), 7.70 (2H, br.m, -NH_2_); ^13^C-NMR: δ 8.73, 32.47, 56.04, 61.85, 104.99, 120.02, 121.07, 125.07, 143.14, 144.08, 146.34, 150.07, 160.93, 197.55; HRMS (*m/z*): calculated value for [M + Na] 100% = 300.0842, found 100% = 300.0842^+^. Anal. Calc. for C_14_H_15_NO_5_: 60.64% C; 5.45% H, 5.05% N, found 60.04% C; 4.537% H, 4.81% N.

##### Synthesis of Methyl 7-Acetyl-5,6-Dimethoxy-3-Methylbenzofuran-2-Carboxylate (**1b**)

Methyl 7-acetyl-5,6-dimethoxy-3-methyl-1-benzofuran-2-carboxylate was obtained according to Procedure 2. M.W. = 292.2839; C_15_H_16_O_6_; Yield: 60%; white powder, m.p. 98–100 °C; ^1^H-NMR (300 MHz, CDCl_3_, δ/ppm): 2.55 (3H, s, -CH_3_), 2.73 (3H, s, -COCH_3_), 3.94 (3H, s, -OCH_3_), 3.95 (3H, s, -OCH_3_), 3.95 (3H, s, -COOCH_3_), 7.10 (1H, s, Ar-H); ^13^C-NMR: δ 9.22, 32.34, 51.84, 56.38, 62.40, 104.42, 120.28, 124.84, 125.43, 141.68, 145.57, 148.67, 150.58, 160.41, 197.50; HRMS (*m/z*): calculated value for [M + Na] 100% = 315.0839, found 100% = 315.0839. Anal. Calc. for C_15_H_16_O_6_: 61.64% C; 5.52% H, found 61.35% C; 5.516% H.

##### Synthesis of Methyl 7-Acetyl-3-(Bromomethyl)-5,6-Dimethoxybenzofuran-2-Carboxylate (**1c**) 

Methyl 7-acetyl-3-(bromomethyl)-5,6-dimethoxybenzofuran-2-carboxylate was obtained according to Procedure 4 (method a). M.W. = 371.1800; C_15_H_15_BrO_6_; Yield: 20%; white powder, m.p. 124–125 °C; ^1^H-NMR (300 MHz, CDCl_3_, δ/ppm): 2.72 (3H, s, -COCH_3_), 3.97 (9H, m, -OCH_3_, -OCH_3_, -COOCH_3_), 4.90 (2H, -CH_2_Br), 7.33 (1H, s, Ar-H); ^13^C-NMR: δ 20.46, 32.30, 52.35, 56.43, 62.41, 104.32, 120.53, 122.56, 124.99, 141.66, 145.71, 149.11, 151.01, 159.61, 197.15; HRMS (*m/z*): calculated value for [M + Na] 100% = 392.9944, 99% = 394.9927, found 100% = 392.9945, 99% = 394.9926. Anal. Calc. for C_15_H_14_ BrO_6_: 48.54% C; 4.07% H, found 48.82% C; 4.15% H.

##### Synthesis of 1-(2-Bromo-5,6-Dimethoxy-3-Methylbenzofuran-7-yl)ethanone (**1d**)

1-(2-Bromo-5,6-dimethoxy-3-methylbenzofuran-7-yl)ethanone was obtained according to Procedure 3 (in ethanol) as well as Procedure 4 (method b). Yield: M.W. = 313.1439; C_13_H_13_BrO_4_; Yield: 27%; white powder, m.p. 94–95 °C; ^1^H-NMR (300 MHz, CDCl_3_, δ/ppm): 2.16 (3H, s, -CH_3_), 2.69 (3H, s, -COCH_3_), 3.90 (3H, s, -OCH_3_), 3.93 (3H, s, -OCH_3_), 6.98 (1H, s, Ar-H); ^13^C-NMR: δ 8.69, 32.26, 56.47, 62.38, 103.39, 114.99, 119.83, 125.37, 126.39, 145.20, 1454.74, 150.22, 197.82; HRMS (*m/z*): calculated value for [M + Na] 100% = 334.9889, 99% = 336.9870, found 100% = 334.9897, 99% = 336.9869. Anal. Calc. for C_13_H_13_BrO_4_: 49.86% C; 4.18% H, found 49.76% C; 4.159% H.

##### Synthesis of 2-Bromo-1-(2-Bromo-5,6-Dimethoxy-3-Methylbenzofuran-7-yl]ethanone (**1e**)

2-Bromo-1-(2-bromo-5,6-dimethoxy-3-methylbenzofuran-7-yl)ethanone was obtained according to Procedure 3 (in ethanol). M.W. = 392.0399; C_13_H_12_Br_2_O_4_; Yield: 27%; white powder, m.p. 128–129 °C; ^1^H-NMR (300 MHz, CDCl_3_, δ/ppm): 2.17 (3H, s, -CH_3_), 3.94 (6H, s, -OCH_3,_ -OCH_3_), 4.58 (2H, s, -COCH_2_Br), 7.03 (1H, s, Ar-H); ^13^C-NMR: δ 8.69, 36.49, 56.53, 62.49, 104.60, 115.16, 116.09, 125.46, 126.66, 145.84, 145.95, 150.10, 190.29; HRMS (*m/z*): calculated value for [M + Na] 50% = 412.8995, 100% = 414.8975, 49% = 416.8956, found 50% = 412.8991, 100% = 414.8973, 49% = 416.8944. Anal. Calc. for C_13_H_12_ Br_2_O_4_: 39.83% C; 3.09% H, found 40.21% C; 3.066% H.

#### 4.2.2. Synthesis of Analogues of Compound **IX**

##### Synthesis of 6-Acetyl-5-Hydroxy-2-Methylbenzofuran-3-Carboxamide (**2a**)

6-Acetyl-5-hydroxy-2-methylbenzofuran-3-carboxamide was obtained according to Procedure 1. M.W. = 233.2200; C_12_H_11_NO_4_; Yield: 50%; white powder, m.p. 268–267 °C; ^1^H-NMR (300 MHz, DMSO, δ/ppm): 2.65 (3H, s, -COCH_3_), 2.68 (3H, s, -CH_3_), 7.22 (1H, s, Ar-H), 7.48 (2H, br.s, -NH_2_), 8.11 (1H, s, Ar-H), 11.97 (1H, s, -OH); ^13^C-NMR: δ 14.20, 27.52, 107.03, 112.69, 112.74, 116.38, 133.39, 146.00, 157.39, 163.86, 164.22, 204.26; HRMS (*m/z*): calculated value for [M + Na] 100% = 256.0580, found 100% = 256.0582. Anal. Calc. for C_12_H_11_NO_4_: 61.80% C; 4.75% H, 6.01% N, found 60.85% C; 4.766% H, 5.83% N.

##### Synthesis of 6-Acetyl-4-Bromo-5-Hydroxy-2-Methylbenzofuran-3-Carboxamide (**2b**)

6-Acetyl-4-bromo-5-hydroxy-2-methylbenzofuran-3-carboxamide was obtained according to Procedure 4 (method b). M.W. = 312.1161; C_12_H_10_BrNO_4_; Yield: 40%; white powder, m.p. 249–248 °C; ^1^H-NMR (300 MHz, DMSO, δ/ppm): 2.50 (3H, s, -COCH_3_), 2.68 (3H, s, -CH_3_), 7.69 (1H, br.s, -NH_2_), 7.94 (1H, br.s, -NH_2_), 8.23 (1H, s, Ar-H), 12.88 (1H, s, -OH); ^13^C-NMR: δ 13.20, 27.05, 99.21, 112.59, 115.82, 115.86, 133.19, 145.62, 154.37, 160.33, 163.79, 204.99; HRMS (*m/z*): calculated value for [M + Na] 100% = 333.9685, 98% = 335.9666, found 100% = 333.9685, 98% = 335.9669. Anal. Calc. for C_12_H_10_BrNO_4_: 46.18% C; 3.23% H, 4.49% N, found 46.33% C; 3.265% H, 4.35% N.

##### Synthesis of Methyl 6-Acetyl-5-Hydroxy-2-Methylbenzofuran-3-Carboxylate (**2c**) 

Methyl 6-acetyl-5-hydroxy-2-methylbenzofuran-3-carboxylate (**2c**) was obtained according to the method described previously [11].

##### Synthesis of Methyl 6-Acetyl-2-(Bromomethyl)-5-Hydroxybenzofuran-3-Carboxylate (**2d**)

Methyl 6-acetyl-2-(bromomethyl)-5-hydroxybenzofuran-3-carboxylate was obtained according to Procedure 3 (in CCl_4_). M.W. = 327.1274; C_13_H_11_BrO_5_; Yield: 30%; white powder, m.p. 94–95 °C (chyba 138–140); ^1^H-NMR (300 MHz, CDCl_3_, δ/ppm): 2.70 (3H, s, -COCH_3_), 3.99 (3H, s, -COOCH_3_), 4.94 (2H, s, -CH_2_Br), 7.53 (1H, s, Ar-H), 7.88 (1H, s, Ar-H), 12.11 (1H, s, -OH); ^13^C-NMR: δ 14.19, 26.80, 52.08, 110.51, 112.63, 117.59, 132.64, 146.73, 155.99, 161.21, 162.63, 164.00, 203.35; HRMS (*m/z*): calculated value for [M + Na] 100% = 348.9682, 99% = 350.9663, found 100% = 348.9683, 99% = 350.9663. Anal. Calc. for C_13_H_11_BrO_5_: 47.73% C; 3.39% H, found 47.33% C; 3.265% H. 

##### Synthesis of 6-Acetyl-4-Bromo-5-Hydroxy-2-Methylbenzofuran-3-Carboxylic Acid (**2e**)

6-Acetyl-4-bromo-5-hydroxy-2-methylbenzofuran-3-carboxylic acid was obtained according Procedure 4 (method b). M.W. = 313.1008; C_12_H_9_BrO_5_; Yield: 30%; white powder, m.p. 196–197 °C; ^1^H-NMR (300 MHz, CDCl_3_, δ/ppm): 2.48 (3H, s, -COCH_3_), 2.69 (3H, s, -CH_3_), 7.72 (1H, s, Ar-H), 12.89 (1H, s, -OH); ^13^C-NMR: δ 13.20, 26.70, 94.45, 100.68, 111.33, 116.11, 132.08, 145.99, 155.30, 159.72, 203.35; HRMS (*m/z*): calculated value for [M + Na] 100% = 334.9526, 99% = 336.9506, found 100%= 334.9525, 99% = 336.9505. Anal. Calc. for C_12_H_9_BrO_5_: 46.03% C; 2.90% H, found 46.33% C; 2.265% H. 

#### 4.2.3. Synthesis of Analogues of Compound **X**

##### Synthesis of Methyl 6-Acetyl-5-Methoxy-2-Methylbenzofuran-3-Carboxylate (**3**) and Methyl 6-(Dibromoacetyl)-5-Methoxy-2-Methyl-1-Benzofuran-3-Carboxylate (**X**) 

Methyl 6-acetyl-5-methoxy-2-methylbenzofuran-3-carboxylate (**3**) and methyl 6-(dibromoacetyl)-5-methoxy-2-methylbenzofuran-3-carboxylate (**X**) were obtained according to the method described previously [11].

##### Synthesis of Methyl 6-(2,2-Dibromo-1-Hydroxyethyl)-5-Methoxy-2-Methylbenzofuran-3-Carboxylate (**3a**)

Methyl 6-(2,2-dibromo-1-hydroxyethyl)-5-methoxy-2-methylbenzofuran-3-carboxylate (**3a**) was obtained according to Procedure 5. M.W. = 422.0659; C_14_H_14_Br_2_O_5_; Yield: 70%; white powder, m.p. 170–172 °C; ^1^H-NMR (300 MHz, DMSO, δ/ppm): 2.74 (3H, s, -COOCH_3_), 3.18 (1H, br.s, -CH-), 3.93 (3H, s, -OCH_3_), 3.94 (3H, s, -CH_3_), 5.34 (1H, br.s, –OH), 6.14 (1H, d, -CH-, *J* = 3 Hz), 7.41 (1H, s, Ar-H), 7.59 (1H, s, Ar-H); ^13^C-NMR: δ 14.73, 51.46, 51.77, 55.93, 75.00, 102.26, 108.96, 110.78, 123.97, 126.92, 148.14, 153.17, 164.71, 164.75; HRMS (*m*/*z*): calculated value for [M + Na] 50% = 442.9100, 100% = 444.9081, 50% = 446.9063, found 50% = 442.9094, 100% = 444.9079, 50% = 446.9065. Anal. Calc. for C_14_H_14_Br_2_O_6_: 39.84% C; 3.34% H, found 40.20% C; 3.37% H. 

##### Synthesis of 6-Acetyl-5-Methoxy-2-Methylbenzofuran-3-Carboxylic Acid (**3b**)

A mixture of methyl 6-acetyl-5-methoxy-2-methylbenzofuran-3-carboxylate (0.0008 mol) and 2 M NaOH (0.6 mL, 0.0012 mol) in ethanol (1.2 mL) was heated for 1 h. The bulk of the solvent was evaporated and the residue was acidified with 2 M HCl (1.2 mL) to give a fine precipitate. Next, the mixture was cooled to room temperature and then filtered to give the product. M.W. = 248.2313; C_13_H_12_O_5_; Yield: 30%; white powder, m.p. 249–250 °C; ^1^H-NMR (300 MHz, DMSO, δ/ppm): 2.55 (3H, s, -COCH_3_), 2.71 (3H, s, -CH_3_), 3.90 (3H, s, -OCH_3_), 7.45 (1H, s, Ar-H), 7.69 (1H, s, Ar-H); ^13^C-NMR: δ 14.41, 31.61, 56.03, 103.27, 109.23, 111.32, 124.90, 130.79, 147.06, 155.84, 164.66, 166.58, 197.99; HRMS (*m*/*z*): calculated value for [M + Na] 100% = 271.0577, found 100% = 270.0737. Anal. Calc. for C_13_H_12_O_5_* ½ H_2_O: 60.50% C; 5.07% H, found 60.05% C; 4.715% H. 

##### Synthesis of 6-Acetyl-5-Methoxy-2-Methylbenzofuran-3-Carboxamide (**3c**)

6-Acetyl-5-methoxy-2-methylbenzofuran-3-carboxamide was obtained according to Procedure 1.

M.W. = 247.2466; C_13_H_13_NO_4_; Yield: 30%; white powder, m.p. 223–224 °C; ^1^H-NMR (300 MHz, DMSO, δ/ppm): 2.55 (3H, s, -COCH_3_), 2.68 (3H, s, -CH_3_), 3.98 (3H, s, -OCH_3_), 7.22 (1H, s, Ar-H), 7.53 (2H, br.s, -NH_2_), 7.70 (1H, s, Ar-H); ^13^C-NMR: δ 14.11, 31.61, 56.17, 103.10, 111.14, 112.81, 124.65, 130.56, 146.94, 155.49, 162.02, 164.42, 198.08; HRMS (*m/z*): calculated value for [M + Na] 100% = 270.0737, found 100% = 270.0737. Anal. Calc. for C_13_H_13_NO_4_: 63.15% C; 5.30% H, 5.67% N, found 62.90% C; 5.245% H, 5.63% N. 

##### Synthesis of Methyl 6-Acetyl-2-(Bromomethyl)-5-Methoxybenzofuran-3-Carboxylate (**3d**)

Methyl 6-acetyl-2-(bromomethyl)-5-methoxybenzofuran-3-carboxylate was obtained according to Procedure 3 (in CCl_4_). M.W. = 341.1540; C_14_H_13_BrO_5_; Yield: 30%; white powder, m.p. 148–150 °C; ^1^H-NMR (300 MHz, CDCl_3_, δ/ppm): 2.65 (3H, s, -COCH_3_), 3.98 (3H, s, -OCH_3_), 4.01 (3H, s, -COOCH3), 4.91 (2H, s, -CH2Br), 7.50 (1H, s, Ar-H), 7.86 7.70 (1H, s, Ar-H); ^13^C-NMR: δ 20.94, 31.08, 52.07, 56.03, 103.76, 110.48, 112.98, 127.44, 129.80, 148.49, 156.56, 161.91, 163.30, 199.04; HRMS (*m/z*): calculated value for [M + Na] 100% = 362.9839, 99% = 364.9820, found 100% = 362.9839, 99% = 364.9820. Anal. Calc. for C_14_H_13_BrO_5_: 49.29% C; 3.84% H, found 48.87% C; 3.755% H. 

### 4.3. Anticancer Activity

#### 4.3.1. Cells and Cytotoxicity Assay

Human umbilical vein endothelial cells (Life Technologies, Waltham, MA, USA) were cultured (according to the manufacturer instructions) in Medium 200 supplemented with Low Serum Growth Supplement. 1 × 10^4^ HUVEC cells were seeded on each well on a 96-well plate (Nunc). The HeLa (human cervix carcinoma) K562 and MOLT-4 (leukemia) cells were cultured in RPMI 1640 medium supplemented with antibiotics and 10% fetal calf serum (HeLa, K562) in a 5% CO_2_-95% air atmosphere. 7 × 10^3^ HeLa, K562, or MOLT-4 cells were seeded on each well on 96-well plate (Nunc). 24 h later cells were treated with the test compounds and then incubated for an additional 48 hours. Stock solutions of test compounds were freshly prepared in DMSO (dimethylsulfoxide). The final concentrations of compounds that were tested in the cell cultures were: 2 × 10^−1^, 1 ×10^−1^, 5 × 10^−2^, 1 × 10^−2^, 1 × 10^−3^ and 1 × 10^−4^ mM. The concentration of DMSO in the cell culture medium was 1%.

The values of IC_50_ (the concentration of test compound that is required to reduce the cell survival fraction to 50% of the control) were calculated from dose-response curves and used as a measure of cellular sensitivity to a given treatment.

The MTT [3-(4,5-dimethylthiazol-2-yl)-2,5-diphenyltetrazolium bromide; Sigma, St. Louis, MO, USA] assay determined the cytotoxicity of all the compounds, as described previously [30]. Briefly, after 24 h or 48 h of incubation with the drug, the cells were treated with the MTT reagent, and incubation was continued for 2 h. MTT-formazan crystals were dissolved in 20% SDS and 50% DMF at pH 4.7 and absorbance was read at 570 and 650 nm on an ELISA-PLATE READER (FLUOstar Omega, BMG LABTECH GmbH, Ortenberg, Germany). As a control (100% viability), cells that were grown in the presence of medium vehicle only with 1% DMSO were used.

#### 4.3.2. Induction of Cell Apoptosis Analyzed by Caspase-3/7 Assay

20 × 10^3^ K562 cells were seeded on each well of 96-well plate in RPMI 1640 medium supplemented with 10% fetal calf serum and antibiotics. Cells were grown for 24 h at 37 °C and 5% CO_2_. The test compounds were dissolved in DMSO and added to the cell culture. The concentration of tested benzo[b]furans in cell culture was 5 × IC_50_.

Cells treated with 1% DMSO served as a negative control, while cells incubated with staurosporine (a strong inducer of apoptosis) were used as a positive control. Cells were exposed to test compounds for 18 h at 37 °C and 5% CO_2_. Subsequently, Apo-ONE^®^ Homogeneous Caspase-3/7 Assay (Promega, Madison, WI, USA) measured the activity of caspase 3 and 7, according to the manufacturer’s instructions. Briefly, the cells were lysed and incubated for 1.5 h with profluorescent substrate for caspase 3 and 7. Next, fluorescence was read at an excitation wavelength of 485 nm and emission of 520 nm with FLUOStar Omega plate reader (BMG-Labtech, Ortenberg, Germany).

#### 4.3.3. Digestion of Plasmid DNA with BamHI Restriction Nuclease

0.5 μg of plasmid DNA (pcDNAHisC, total length 5.5 kbp) containing a unique *BamHI* restriction site was dissolved in a 1× *BamHI* reaction buffer and then incubated overnight at 37 °C with the test compounds or daunorubicin, a strong intercalating agent, which was used as a positive control. The concentration of the test compounds and daunorubicin samples was 100 µM. In the next step, the reaction mixtures were digested with *Bam**HI* restriction endonuclease (2 U/μL) for 3 h at 37 °C. The total reaction volume was 10 μL. Products of the reaction were subjected to the 1% agarose gel electrophoresis in TBE buffer. The gel was stained with ethidium bromide and DNA fragments were visualized under a UV lamp (GBox, Syngene, Cambridge, UK).

## 5. Conclusions

We synthesized and tested a group of new benzofuran derivatives. The presence of bromine in the alkyl group in the furan ring is most likely responsible for the cytotoxic properties of the tested derivatives (compounds **1c**, **2d**, **3d**). Compound **1e** shows the cytotoxic property, and contains an acetyl halide substituent (bromine) in the benzene ring and a bromine atom that is directly attached to the furan ring. The most active compounds **2d** and **3d**, showed increased polarity when compared to the lead compounds **VIII**-**X**, but their cytotoxicity against human cancer cells decreased by 5–10 folds and the toxicity against normal cells increased. The formation of amide derivatives of benzofurans (compounds **1a**, **2a**, **2b**, **3c**) and the lack of a halogen-containing alkyl substituent in their structure resulted in better water solubility but loss of cytotoxic properties towards the cancer cells studied. A reduction of the bromoacetyl group in compound **X** increased its polarity but also eliminated the selectivity of the compound and diminished its toxicity towards tumor cells.

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
