# Peer review of "Synthesis of New Derivatives of Benzofuran as Potential Anticancer Agents"

_molecules, 2019, doi:10.3390/molecules24081529_

Round 1

Reviewer 1 Report

The authors should consider at least the following suggestions/corrections/improvements:

1. Line 24: HL60 is referred as a key-word, but HL60 cells have not been used in the biological evaluation of the synthesized compounds. I wonder whether this should be MOLT-4 instead.

2. Figure 1: Three of the five structures in this figure are wrong. These are: conocarpan (no double bond in the heterocyclic ring, opposite stereochemistry, a HO-group at position 4 is missing from the phenyl substituent), ailanthoidol (the methoxy group is ortho- , not meta-, to the HO-group) and amiodarone (the ethyl group should be erased, the two CH3 groups should be replaced by I and between the N and O atom are two CH2 groups, not one). The authors should check the related literature and revise accordingly.

3. A general suggestion is that in all figures the authors should use higher letter size than the one used. The letters are too small for the size of the drawings.

4. Line 122:  .....6-acetyl-5-hydroxy-2-methyl-1-benzofuran-3-carboxylic acid should be replaced by .....6-acetyl-5-hydroxy-2-methylbenzofuran-3-carboxylic acid

5. Line 132:  ...by used... should be replaced .....by using......

6. Line 134: ...confirming better solubility...... should be replaced by ...confirming better water solubility......

7. Line 135: ..... formaldehyde derivatives 2a and 2b…. should be replaced by … primary carboxamides 2a and 2b…..

8. In Table 1, the first compound is not VII but VIII.

9. Line 220: …..14 new derivatives with decreased lipophilicity as evidenced by ClogP….Note: This is not true for compounds 1c-1e compared to lead compound VIII.

10. Line 248-249, .....This effect is probably due to the absence of the bromoacetyl group at the 2-position of the furan ring….Note: there is not such a group at position 2 of furan ring in XIII, but an acetyl group

11. Line 251: the word ‘configuration’ should be better replaced by e.g. orientation or arrangement, as ‘configuration’ refers to stereochemistry

12. Lines 311 and 317: ..amid… should be replaced by ..amide..

13. Lines 334, 342, 364, 372, 380, 387, 406, 437: The names of the compounds should start with a capital letter, e.g. 7-Acetyl-5,6-dimethoxy……

14. From Line 333 to Line 445: In the names of all compounds ‘-1-‘ should be erased, e.g. the correct name for compound (1) should be: 7-acetyl-5,6-dimethoxy-3-methylbenzofuran-2-carboxylic acid

15. From Line 339 and below: Ιn the HRMS data, the calculated value should be provide first, followed by the one found, e.g. for compound (1) the calculated value for [M+Na]+ is 301.0688.

     Also for some of the compounds, namely (1), (1a), (2a), the elemental analysis date for C is not satisfactory, that is within ±0.4%.

16. Line 437, 6-acetyl-5-methoxy-2-methyl-1-benzofuran-3-carboxamide (BM6HA): it is not clear what does BM6HA mean or stands for?

17. Line 495: Conclusions. Note: The main aim of the present work seems to be the improvement of the poor water solubility of lead compounds XIII-X (see lines 80-86). Therefore, the authors should discuss briefly in the conclusion section whether this was achieved or not with the synthesized compounds and what was the effect of reducing lipophilicity on the anti-cancer activity of the synthesized compounds. Furthermore, to compare the cytotoxicity of the new compounds against the lead ones and the fact that the new compounds were of comparable cytotoxic (with one exception) to the healthy HUVEC cells, in contrast to the lead ones.

18. Line 504: …..2.1 Subsection. This must be erased

19. Reference [30] appears in Line 228 whereas reference [29] appears in Line 468. Therefore, ref. [30] should become ref. [29] and vice versa.

20. Line 515: References. The authors should revise carefully this section so that the presentation of all references is consistent. See some examples below:

     (a) Compare e.g. Phytoter Res (line 517) without full stops and Bioorg. Med. Chem. Lett. (line 522-23) with full stops

     (b) Z. Naturforsc C (line 527). Please use italics and full stop before C

    (c)  Compare Anti-Cancer Agents in Medicinal Chemistry, 2015; 15(1):115-121. (line 570) and Anticancer Agents Med Chem, 2018 Nov 22….. (line 576). The correct should be Anticancer Agents Med. Chem. 2015, 15(1), 115-121 for the former and analogously for the latter.

     (d) Infection and Immunity 2005, 73(4), 1907-1916 (line 584). The correct should be Infect. Immun. 2005, 73(4), 1907-1916.

Author Response

Dear Reviewer,

Thank you for your comments. We have revised our manuscript according to your suggestions. All changes are highlighted using the “Track Changes” function in Microsoft Word. We hope that our revisions are satisfactory and will allow to publish the manuscript. Please, find in the attachment our response to your comments point by point.

Best regards,

Mariola Napiórkowska

Reviewer 2 Report

The manuscript “molecules-484009” described the synthesis and characterization of new analogs of the benzopyran compounds, previously developed by the authors, to increase polarity. The leading compounds, 2d and 3d, showed increased polarity, but their cytotoxicity against human cancer cells decreased by 5-10 folds and toxicity against normal cells increased. The derivatives will not be useful for further anticancer drug development. It was well described in Line 167, 168, 170, and the discussion.

I recommend the publication of the manuscript with minor revisions.

Line 16: “a” should be removed.

Line 19: K562 (cancer type)

Line 21: “cancer” should be removed. “against all cell lines and did not show selectivity for cancer cell lines.”

Line 23: increased their cytotoxicity both in normal and cancer cells.

Line 29: “Both” can be removed and “Natural and…”

Line 31: “immunosuppressive”

Line 53: “induced apoptosis of human HCC cells” to “induced their apoptosis”

Line 54: “effective an” to “an effective”

Line 56: “report” to “reported”

Line 59: “are” to “were”

Line 61: Do “human cell lines” mean “human normal cells” or “human cancer cell lines”?

Line 76: “, however” to “; however,”

Line 76: Do “adherent cancer cells” mean that the physical state (suspension or adhesion) is more important than identity of a certain cancer cell line? _its clarification is needed.

Line 83: “VIII-XI” to “VIII-X”

Line 95: The meaning is confusing because the lead compound VIII has a bromomethyl group in the position 3.

Line 102: “the analog of VIII” to “1c”

Line 134: “logP” to “clogP” unless the value was measured experimentally.

Line 135: “formaldehyde derivatives” to “amide derivatives”

Line 143, 154, and 156: “logP” to “clogP”

Line 154: “leader” to “lead”

Line 167: “not shown” to “data not shown”

Figure 5: “K562” should be replaced by “untreated” or “negative control”

Line 220: “with decreased ClogP values” The ClogP value will indicate the polarity trends but cannot be an evidence.

Line 285: “05.” to “0.5”

Line 504 “2.1. subsection”

Author Response

Dear Reviewer,

Thank you for your comments. We have revised our manuscript according to your suggestions. All changes are highlighted using the “Track Changes” function in Microsoft Word. We hope that our revisions are satisfactory and will allow to publish the manuscript. Please, find in the attachment our response to your comments point by point.

Best regards

Mariola Napiórkowska
